# Novel Methods of Targeting IL-1 Signalling for the Treatment of Breast Cancer Bone Metastasis

**DOI:** 10.3390/cancers14194816

**Published:** 2022-10-01

**Authors:** Jiabao Zhou, Jennifer M. Down, Christopher N. George, Jessica Murphy, Diane V. Lefley, Claudia Tulotta, Marwa A. Alsharif, Michael Leach, Penelope D. Ottewell

**Affiliations:** 1Academic Unit of Clinical Oncology, Department of Oncology and Metabolism, Medical School, University of Sheffield, Sheffield S10 2RX, UK; 2Institute of Experimental Pathology (ExPath), Center of Molecular Biology of Inflammation (ZMBE), Westfälische Wilhelms-Universität (WWU), 48149 Münster, Germany; 3School of Science, University of Greenwich at Medway, Chatham ME4 4TB, UK

**Keywords:** breast cancer, bone metastasis, IL1β, mouse models

## Abstract

**Simple Summary:**

The pro-inflammatory cytokine, IL1β, plays a pivotal role in breast cancer bone metastasis. Inhibiting IL-1 signalling with the IL1β specific antibody, Canakinumab, or the IL1R1 antagonist Anakinra almost eliminates bone metastases but has adverse effects on tumours growing outside of the bone and immune regulation. This current study demonstrated that pharmacological inhibition of other members of the IL-1 signalling pathway Caspase-1, IL1β and IL1R reduced migration and invasion of E0771 and Py8119 cells in vitro and also reduced spontaneous metastasis and metastatic outgrowth of breast cancer in the bone, in vivo. Interestingly, targeting IRAK1 had no anti-tumour effects. Importantly, inhibiting Caspase-1 reduces bone metastasis without adversely affecting tumours outside of bone or immune cell regulation, suggesting that targeting immediately upstream of IL1β may be a good therapeutic strategy for treating patients with breast-cancer-induced bone disease.

**Abstract:**

Breast cancer bone metastasis is currently incurable. Evidence suggests that inhibiting IL-1 signalling with the IL1R antagonist, Anakinra, or the IL1β antibody, Canakinumab, prevents metastasis and almost eliminates breast cancer growth in the bone. However, these drugs increase primary tumour growth. We, therefore, investigated whether targeting other members of the IL-1 pathway (Caspase-1, IL1β or IRAK1) could reduce bone metastases without increasing tumour growth outside of the bone. Inhibition of IL-1 via MLX01 (IL1β secretion inhibitor), VRT043198/VX765 (Caspase-1 inhibitor), Pacritinib (IRAK1 inhibitor) or Anakinra (IL1R antagonist) on tumour cell viability, migration and invasion were assessed in mouse mammary E0771 and Py8119 cells in vitro and on primary tumour growth, spontaneous metastasis and metastatic outgrowth in vivo. In vitro, Inhibition of IL-1 signalling by MLX01, VRT043198 and Anakinra reduced migration of E0771 and Py8119 cells and reversed tumour-derived IL1β induced-increased invasion and migration towards bone cells. In vivo, VX765 and Anakinra significantly reduced spontaneous metastasis and metastatic outgrowth in the bone, whereas MLX01 reduced primary tumour growth and bone metastasis. Pacritinib had no effect on metastasis in vitro or in vivo. Targeting IL-1 signalling with small molecule inhibitors may provide a new therapeutic strategy for breast cancer bone metastasis.

## 1. Introduction

Over 2.3 million women are diagnosed with breast cancer every year, and ~685,000 will die as a result of this condition (www.who.int) (accessed on 23 May 2022). Whilst improvements in targeted treatments for primary breast cancers have proven to be beneficial, with >80% of women making a full recovery, for patients whose tumours spread to distant organs, there are still no curative treatments. Around 70–80% of patients with late-stage breast cancer develop secondary tumours in their skeletons, and life expectancy for these patients drops to 2–3 years following diagnosis of bone involvement [1]. There is, therefore, a clear need for improved therapies for patients whose tumours spread to bones. There is increasing evidence that the pro-inflammatory cytokine IL1β plays a critical role in breast cancer spread to the bone and metastatic outgrowth in this site: In patient samples, expression of IL1β by breast cancer cells is associated with an increased likelihood of future relapse in the bone [2,3]. Overexpression of IL1β in tumour cells facilitates epithelial to mesenchymal transition (EMT), migration and invasion towards bone cells in vitro, as well as inducing bone homing properties and promoting the dissemination of tumour cells in the bone in in vivo models [2,3,4,5,6,7,8]. Once cancer cells are disseminated in the bone, interactions between tumour cells and the bone metastatic niche (osteoblasts, haematopoietic stem cells and blood vessels) lead to increased secretion of IL1β in the local environment. Environmental IL1β promotes the expansion of the niche, and IL1β driven wnt activation leads to metastatic outgrowth of tumour cells disseminated in this site [2,7,9,10]. Importantly, inhibition of IL-1 signalling with the anti-IL1β antibody Canakinumab, or the IL1R antagonist, Anakinra, inhibits bone metastasis in pre-clinical models of breast cancer. Similarly, Canakinumab has been shown to have potent anti-cancer effects in a clinical trial [11], suggesting that targeting IL-1 may be an attractive therapeutic strategy for preventing/treating breast cancer-induced bone disease 1, [12,13].

In addition to its direct anti-metastatic properties, IL1β plays a central role in regulating immunity [14], and it has recently become apparent that IL1β within the tumour microenvironment is important for recruiting anti-tumour immune cells, especially in soft tissue tumours [7,10]. Global removal of IL1β in IL1β knockout mice leads to increased growth of primary mammary tumours, and administration of Anakinra/Canakinumab also increases the growth of primary breast cancers in mice [2,13]. Anakinra also increases the growth of disseminated tumour cells in the lungs [15,16,17]. Interestingly, IL1β appears to regulate anti-tumour immunity differently in the bone and soft tissue sites, promoting the recruitment of anti-tumour macrophages, neutrophils and Th1 cells in soft tissues and reducing myeloid cells in the bone [13,18]. Moreover, the administration of Canakinumab had been shown to significantly reduce the development of lung cancers in patients with atherosclerosis but had the adverse effect of increasing incidences of serious infections [11,19].

Given the promising outcomes of inhibiting IL-1 in suppressing breast cancer bone metastasis, we have evaluated the effects of small molecule inhibitors that target different parts of the IL1β signalling pathway (Caspase-1, IL1β and IRAK1, Interleukin 1 Receptor Associated Kinase 1) on breast cancer bone metastasis. Using in vitro and in vivo models, we have attempted to identify these inhibitors that could prevent IL1β-driven bone metastasis without stimulating tumour growth outside the bone or adversely affecting systemic immunity.

## 2. Materials and Methods

### 2.1. Cell Lines

Mouse mammary tumour E0771 were originally provided by Professor Sandra McAllister’s laboratory (Department of Medicine, Harvard Medical School, Boston, MA, USA) and Py8119 cells purchased from ATCC. These cells were manipulated to make E0771 Luc2 V5 GFP, E0771 Luc2 V5 IL1B GFP, E0771 Luc2 V5 IL1R1 cell lines and Py8119 Luc2 V5 GFP, Py8119 Luc2 V5 IL1B GFP, and Py8119 Luc2 V5 IL1R1 GFP as previously published and expression levels of IL1B and IL1R1 were confirmed by real-time PCR and ELISA (Appendix A) [13]. E0771 and Py8119 Cell lines were cultured in RPMI 1640 Medium (61870-010, Gibco, Waltham, MA, USA) with 10% *v*/*v* fetal bovine serum (FBS) (F7524, Sigma, St. Louis, MI, USA). Pre-osteoblast OB1 cells were a kind gift from Marianna Kruithof de Julio (University of Bern, Bern, Switzerland), and this cell was cultured by DMEM medium (61966-021, Gibco) with 10% *v*/*v* FBS. HUVECs were purchased from PromoCell (Heidelberg, Germany) and cultured in the media supplied (Promocell, Heidelberg, Germany, C-12211 and C-39220).

### 2.2. Drugs

Anakinra (r-metHu1L-ra) was obtained from Amgen, Cambridge, UK. MLX01 was obtained from Michael Leach University of Greenwich, and the evidence of MLX01 inhibition of IL1β production in human cell lines is shown in Appendix A, and in mouse tissue is shown in Appendix A. VX765 (S2228) was obtained from Selleckchem (Boston, MA, USA). VRT043198 (244133-31-1) and Pacritinib (202571) were purchased from Medkoo (Morrisville, NC, USA). Recombinant Mouse IL1β (575106) was purchased from BioLegend (San Diego, CA, USA).

### 2.3. In Vitro Studies

MTT: Cell viability was assessed by MTT according to the manufacturer’s instruction (M5655-1G, Sigma-Aldrich, St. Louis, MI, USA), and the plate was finally read at 570 nm by FlexStation3 Benchtop Multi-Mode Microplate Reader (Molecular Devices, San Jose, CA, USA). Scratch assay: Cell migration was assessed by scratch assay: 100% confluent monolayers of E0771 or Py8119 cells were treated with 10 µg/mL Mitomycin C (M-0503, Sigma-Aldrich) for 3 hours (Mitomycin C serves to inhibit cell proliferation from eliminating the impact of cell division on data analysis from scratch assay and transwell assays.) in a 12-well plate before a scratch was made by a 200µL-tip. Six different fields in each scratch were photographed by EVOS™ FL Auto 2 Imaging System (Fisher Scientific, Waltham, MA, USA) at 0 h, 24 h, and 48 h post the scratch and finally, T-scratch software (ETH Zurich, Zürich, Switzerland) [20] was used to analyse % of open area in the image. Transwell assay: Two types of transwell inserts were respectively used to assess the migratory and invasive ability of E0771 and Py8119 cells towards pre-osteoblast, OB1 cells, Costar transwell permeable Supports 6.5 mm insert with 8um Polycarbonate Membrane (31019046) for migration and Corning BioCoat Matrigel Invasion insert with 8μm Polycarbonate Membrane (354480) for invasion. Firstly, pre-osteoblast OB1 cells or HUVECs were seeded at 30,000 cells/well in a 24-well-plate for 24 h. The next day, a 70% confluent monolayer of E0771/Py8119 cells was treated with 10 µg/mL Mitomycin C (Sigma-Aldrich, M-0503) for 3 hours before cells were collected and resuspended in 0.1% *w*/*v* BSA (A7030, Sigma) RPMI 1640 (FBS free) at 1 × 10^5^ cells/well. A total of 200 μL of this cell suspension was subsequently added into the transwell insert. This insert was suspended above with OB1 cells following addition of 600 μL, fresh, complete DMEM or with HUVECs in 600 μL endothelial cell media supplied with these cells. After 24 h post cell seeding, un-invaded cells from the top of the membrane of the insert were wiped with a cotton bud before the membrane was fixed in 100% ethanol for 5 min and stained with haematoxylin solution (5 min) and 1% *w*/*v* eosin (1 min). The membrane was removed from the insert and mounted onto a glass slide using the aqueous mounting medium (S3025, DAKO, Glostrup, Denmark). Finally, the stained membrane was photographed by Leica microscope with Olympus DP73 colour camera (Tokyo, Japan), and the number of cells was counted.

### 2.4. In Vivo Studies

In vivo procedures were conducted in accordance with local guidelines and with UK Home Office approval under Project License (PPL) 70/8964 or P99922A2E, University of Sheffield, UK. Mice were maintained on a 12 h light:12 h dark cycle with food and water provided ad libitum.

Model of metastatic outgrowth following tumour cell disseminated into metastatic organs (intra-cardiac injection): 2.5 × 10^4^ E0771 luc2 GFP cells were administered via intra-cardiac injection into 5–6-week C57Bl/6J female mice. Five days post tumour cell injection, mice were imaged by IVIS and randomised into groups of equal tumour size. At this point mice received daily intraperitoneal injection of control (5% *v*/*v* DMSO/PBS), MLX01 (50 mg/kg), VX765 (50 mg/kg), Anakinra (1 mg/kg), or Pacritinib (50mg/kg) or orally gavage of control (5% *v*/*v* DMSO/PBS) or MLX01 (100 mg/kg). Model of spontaneous tumour cells metastasis (intra-ductal injection): 1.25 × 10^5^ E0771 luc2 GFP cells were engrafted via injection into the 4th mammary ducts of 5–6-week C57Bl/6J female mice following subcutaneous injection of 0.002 mg Vetergesic. Seven days post tumour cell injection, when tumour volume reached around 0.027 cm^3^, mice were randomised into groups of equal tumour volume, and at this point, mice were given daily intraperitoneal injections of control (5% *v*/*v* DMSO/PBS), MLX01 (50 mg/kg), VX765 (50 mg/kg), Anakinra (1 mg/kg), respectively. Tumours were measured by digital vernier scale every three days, and mice were culled once their primary tumour reached 1 cm^3^.

### 2.5. In Vivo Imaging

Tumours in soft tissue or hind limbs were measured using bioluminescence-based in vivo imaging (IVIS Lumina II in Vivo Imaging System (PerkinElmer, Waltham, MA, USA) and Living Image^®^ 4.5.4 software) 2-, 7-, 15- and 17-days post tumour cell injection, 5 minutes following subcutaneous injection of 6 mg/kg D-Luciferin (Invitrogen, Waltham, MA, USA). Maximum BLI thresholds were set as 5e5 photons, and minimum BLI thresholds were modified across experimental models to allow the visualisation of tumour-derived bioluminescent signals.

### 2.6. Sample Collection

After dissection and IVIS imaging, half of primary tumours, lungs and both tibias were fixed in 4% *w*/*v* PFA/PBS for 48 hours and stored in 70% *v*/*v* ethanol/distilled water for uCT analysis. The other half of the tumours and lungs and both femurs were snap-frozen by immersing in liquid nitrogen and then stored at −80 °C. Following blood collection, serum was collected by centrifugation and stored at −80 °C. Spleens were stored at −80 °C in 10% *v*/*v* DMSO/ FBS for future flow cytometric analysis.

### 2.7. Microcomputed Tomography Imaging

Microcomputed tomography imaging analyses were carried out via a Skyscan 1172 X-ray–computed μCT scanner (Bruker, Billerica, MA, USA) equipped with the X-ray tube (voltage, 49 kV; current, 200 mA) and the 0.5 mm aluminium filter. Pixel size was set to 4.3 µm, and scanning initiated from the top of the proximal tibia as described previously [21].

### 2.8. ELISA

Mouse serum TRAP5b was assessed by MouseTRAP™ (TRAcP 5b) ELISA (SB-TR103), and mouse serum P1NP was assessed by Elabscience P1NP (Procollagen 1 N-Terminal Propeptide) ELISA kit (E-EL-M0233) in accordance with manufacturer instructions. Mouse IL1β was assessed by ELISA MAX™ Deluxe Set Mouse IL1β (432604, BioLegend)

### 2.9. Flow Cytometry

Frozen spleens were quickly defrosted in a 37 °C water bath and then washed in PBS three times to remove DMSO. Deforested spleens were chopped and digested using Liberase digestion solution (Roche, 0540102001) and then filtered using a 100 µm cell strainer to single cells in 1% *v*/*v* FBS/PBS (FACS buffer). Samples were aliquoted and incubated with fluorochrome-conjugated antibodies and the viability dye (Appendix A) for 45 min on ice. Next, the cell pellet was resuspended with 0.5ml 1% *v*/*v* PFA/PBS. Finally, the cell suspension was transferred to a filtered FACS tube on the ice and analysis was performed by BD LSR II flow cytometer and FlowJo software v10.8.0 (Becton, Dickinson and Company, Franklin Lakes, NJ, USA). The gating strategy is shown in Appendix A.

### 2.10. Statistical Analysis

Data analysis was performed using GraphPad Prism 9 (GraphPad Software Inc., San Diego, CA, USA). Data were analysed using unpaired *t*-tests for single timepoints and two-way ANOVA for time course experiments. All data from in vitro experiments are from a minimum of three repeats from three biological experiments, and their graphs represent mean + SD. All in vivo experiments show all individual points, and their graphs represent mean + SD. Statistical significance was defined as * *p* < 0.05, ** *p* < 0.01, *** *p* < 0.001, **** *p* < 0.0001.

## 3. Results

### 3.1. MLX01, VX765, and Anakinra All Inhibit Migration and Invasion of Breast Cancer Cells In Vitro

MLX01 (small molecule inhibitor of IL1β that reduces secretion but not the transcription of IL1β by human cells in vitro (Appendix A) and in mouse tissue in vivo (Appendix A), VRT043198/VX765 (Caspase-1 inhibitors), and Pacritinib (IRAK1 inhibitor) are small molecule inhibitors that block IL-1 signalling by inhibiting IL1β, Caspase-1 and IRAK1, respectively (Figure 1A). We assessed the effects of these inhibitors on parameters associated with tumour growth and metastasis in mouse mammary cancer cell lines E0771 and Py8119 and compared results with the FDA/NICE approved IL1Ra biologics agent Anakinra which has been shown to produce promising effects in blocking breast cancer metastasis to bone in previous studies [2,12,13]. The function of IL-1 signalling on viability and migration and invasion of breast cancer cells was assessed, in vitro, upon IL1B or IL1R1 overexpressing (Appendix A) or treatment with anti-IL-1 signalling. Our data showed that increasing IL-1 signalling via genetic overexpression of IL1β or IL1R1 or by adding recombinant IL1β did not significantly affect the viability of E0771 and Py8119 (Figure 1B,C and Appendix A). Similarly, pharmacological inhibition of IL1β secretion with MLX01, or inhibition of Caspase-1 with 25 ng/mL VRT043198, inhibition of IL1R1 with Anakinra had no significant effect on tumour cell viability (Figure 1C and Appendix A). Interestingly, inhibition of IRAK1 with 250 pg/mL Pacritinib significantly reduced viability of E0771 and Py8119 (*p* = 0.04, *p* = 0.0247) (Figure 1C and Appendix A). These results demonstrate that alteration of IL-1 by changing IL1β or inhibiting the IL-1 binding with IL1R1 has no effect on the viability of breast cancer cells in vitro.

Our previous study showed that tumour-derived IL1β was much more potent at promoting the migration of MDA-MB-231 and MCF7 breast cancer cells compared with IL1β from the microenvironment [2]. To validate this result in the cell lines used for the current study, we genetically manipulated E0771 and Py8119 cells to increase the expression of IL1β, or IL1R1 and used scratch assays to compare their migratory ability with control cells as well as control cells stimulated with recombinant mouse IL1β (microenvironment derived IL1β). We found that 5 ng/mL mouse recombinant IL1β did not promote the migration of breast cancer cells, while genetic overexpression of IL1β increased the migration of E0771 (*p* = 0.003) and Py8119 cells (*p* = 0.0776) (Figure 2A). Moreover, genetic overexpression of IL1R1 had no effect on the migration of either cell line (Figure 2A). Importantly pharmacological inhibition of IL-1 signalling using 12 ng/mL MLX01 (*p* = 0.0245, *p* = 0.0103), 25 ng/mL VRT043198 (*p* = 0.0028, *p* = 0.0093) or 0.1 mg/mL Anakinra (*p* = 0.0040, *p* = 0.0081) significantly decreased migration of both E0771 and Py8119 cells (Figure 2B). Since Pacritinib significantly reduced cell viability and detached adherent cells from plates, scratch assays were not used to assess the effects of migration of E0771 and Py8119 for Pacritinib.

In order to elucidate the effect of IL-1 signalling on breast cancer cell migration and invasion towards the bone, E0771 and Py8119 cells were seeded into the upper chamber of a transwell assay in which the lower chamber had been pre-seeded with pre-osteoblast OB1 cells. The number of tumour cells migrating through pores in the dividing membrane or invading through a Matrigel-coated membrane was counted after 24 h of co-culture. Overexpression of IL1β significantly increased E0771 and Py8119 cell migration and invasion towards OB1 cells (migration *p* = 0.0026, *p* = 0.0094; invasion *p* = 0.0007, *p* = 0.0001, respectively) (Figure 2C,D), suggesting that tumour derived IL1β enhances the functional properties of breast cancer cells enabling migration and invasion towards bone. These enhanced migratory and invasive abilities were significantly reversed by pharmacological inhibition of IL-1 secretion with 12 ng/mL MLX01 (migration *p* = 0.0858, *p* = 0.0159; invasion *p* = 0.0374, *p* = 0.0482), 25 ng/mL VRT043198 (migration *p* = 0.0317, *p* = 0.0121; invasion *p* = 0.0007, *p* = 0.0232) and 0.1 mg/mL Anakinra (migration *p* = 0.0165, *p* = 0.0467; invasion *p* = 0.001, *p* = 0.0088) (Figure 2C,D). However, inhibition of IRAK1 via 250 pg/mL Pacritinib had no effect on migration or invasion of E0771 IL1β overexpression cells to bone (Appendix A). Moreover, we also investigated the effect of IL1β on invasive ability in the presence of human umbilical vein endothelial cells (to represent the metastatic lung niche, which is comprised of endothelial cells). Interestingly, we found that tumour-derived IL1β decreased the ability of breast cancer cells to invade HUVEC cells (E0771, 0 = 0.0101; Py8119, *p* = 0.0231), which is opposite to the pro-metastatic effects observed toward the bone metastatic niche. Furthermore, pharmacological inhibition of IL-1 signalling (via 12 ng/mL MLX01: Py8119, *p* = 0.0003; 25 ng/mL VRT043198: *p* = 0.0283; and 0.1 mg/mL Anakinra) increased invasion of breast cancer cells to HUVEC cells (Appendix A). These results demonstrate that IL1 signalling in breast cancer cells has different effects on metastasis to different metastatic niches; IL-1 is strongly pro-metastatic towards bone cells but anti-metastatic towards endothelial cells.

### 3.2. MLX01, VX765, and Anakinra Reduce Metastatic Outgrowth of Tumour Cells Disseminated in the Bone In Vivo

To investigate the effects of the small molecule inhibitors of the IL1β signalling pathway, MLX01, VX765 and Pacritinib, on disseminated breast cancer metastasis to bone in vivo, E0771 Luc GFP bone homing cells were injected into the blood circulation of C57BL/6J mice via intra-cardiac injection, and metastatic outgrowth was compared with control (PBS treated mice) and mice treated with Anakinra. Three days post-injection of tumour cells, mice were randomised into the following groups: PBS (control), 50 mg/kg/day MLX01, 50 mg/kg/day VX765, and 1 mg/kg/day Anakinra; all treatments were administered by i.p injection for 13 consecutive days (Figure 3A). Firstly, 50 mg/kg/day MLX01, 50 mg/kg/day VX765 and 1 mg/kg/day Anakinra significantly reduced the number of overt metastasis per mouse (*p* = 0.009, *p* < 0.0001, *p* < 0.0001, respectively) (Figure 3C), whereas administration of 50 mg/kg Pacritinib had no significant effects (Figure 3J). Five days after injection of tumour cells, the development of bone metastases in the hind limbs was reduced upon treatment with MLX01, VX765 and Anakinra (femur/tibia of 6/11 (54.5%) mice in the control group; 1/10 (10%) mice in the MLX01 treatment group; 0/11 (0%) mice in the VX765 treatment group and 0/11 (0%) of mice following treatment with Anakinra) (Figure 3B,E). Sixteen days post tumour injection, numbers of mice with bone metastasis increased to 8/11 (72.7%) of mice in the saline group and 5/10 (50%) of mice in the MLX01 treatment group; however, bone metastases did not develop following administration of VX765 (0/11 (0%) of mice), and only 1/11 (9%) mice developed bone metastasis following Anakinra (Figure 3E and Appendix A). Because FDA/NICE approved IL1β inhibitors, Canakinumab and Anakinra, have poor oral bioavailability making administration difficult for patients (subcutaneous injection), we also assessed the ability of the orally available IL1β inhibitor, MLX01, to inhibit the metastatic outgrowth of breast tumours following oral administration (Appendix A). Because MLX01 reduces IL-1B secretion but not transcription (Appendix A), we used ELISA (Appendix A) to first confirm that oral gavage of 100 mg/kg MLX01 reduced IL-1B secretion in mice to the same level as administering 50 mg/kg via i.p injection (Appendix A) Oral gavage of 100mg/kg MLX01 reduced the number of overt metastases of whole mice (*p* = 0.0085) and reduced metastatic outgrowth of E0771 cells in the bone by 50% and 34%, 10 days and 16 days after tumour cell injection, respectively (Appendix A). Unlike other inhibitors of the IL-1 pathway, inhibition of IRAK1 with Pacritinib had very little effect on bone metastasis, reducing metastatic outgrowth of disseminated tumour cells in the bone by 12% compared with control, 18 days post tumour injection (Figure 3L,I and Appendix A). Interestingly, bioluminescence imaging showed that when tumours did grow in the bone and multiple organs, their volume was not altered following treatment with any inhibitors tested, suggesting that inhibition of the IL1β signalling pathway prevents the initial stages of metastatic outgrowth of disseminated tumour cells in the bone but does not slow down the growth of established tumours in this site (Figure 3D,F,K,M and Appendix A).

Previously published data have shown that inhibiting IL1β signalling with Canakinumab increases the growth of soft tissue metastasis [2]. We, therefore, investigated the effects of small molecule inhibitors of the IL1β signalling pathways on the metastatic outgrowth of E0771 cells disseminated in soft tissues. When we quantified the number of mice in which soft tissue metastases developed, we could see that MLX01, VX765 and Anakinra reduced the percentage of mice with liver metastases by 56%, 60% and 60%, respectively, and kidney metastases by 52%, 85% and 85%, respectively, compared with control mice treated with PBS (Figure 3G). MLX01 treatment, however, increased lung metastasis by 61% and the same effect was confirmed in mice which received 100 mg/kg MLX01 by oral gavage (Figure 3G and Appendix A). Moreover, although VX765 and Anakinra did not increase the percentage of mice with lung metastasis, similar to MLX01, in lungs where metastases did develop, tumour growth was increased. (Appendix A). Pacritinib, on the other hand, exerted no significant effects on soft tissue metastasis (Figure 3N and Appendix A).

### 3.3. MLX01, VX765, and Anakinra Reduce Spontaneous Breast Cancer Metastasis to Bone

Because inhibiting the IL1β signalling pathway with MLX01, VX765 and Anakinra reduced migration and invasion, in vitro, and metastatic outgrowth of disseminated breast cancer cells in the bone, in vivo, we investigated whether administrating these treatments earlier in cancer progression could reduce spontaneous metastasis. For these experiments, E0771 Luc GFP bone homing cells were orthotopically engrafted in C57BL/6J female mice. Seven days later, mice were randomised to the following groups: PBS (control), 50 mg/kg/day MLX01, 50 mg/kg/day VX765 or 1 mg/kg/day Anakinra. Mice were culled on an individual basis once their tumours reached 1000 mm^3^ and assessed for primary tumour growth and development of metastasis (Figure 4A). Our data showed that MLX01 significantly decreased growth of primary tumours compared to saline (*p* < 0.0001) resulting in increased survival (*p* = 0.0199) (Figure 4B,C). However, VX765 and Anakinra had no effect on E0771 tumour growth in the primary site compared with control (Figure 4B). All inhibitors tested had a reduced number of mice that developed spontaneous metastasis to bone: MLX01 reduced bone metastasis by 30%, VX765 reduced bone metastasis by 47%, and Anakinra reduced bone metastases by 41% compared with control (Figure 4D,E). However, in tumours that did grow in the bone, tumour volume was not altered (Figure 4D,E). In addition, MLX01 and Anakinra slightly reduced spontaneous metastasis to the lung compared with control (by 22% and 18%, respectively) but tumours in the lungs of mice treated with VX765 (*p* = 0.0142) and Anakinra (*p* = 0.0072) had higher luminescence levels (Figure 4F,G). These results further suggest inhibition of IL-1 reduces spontaneous metastases and metastatic outgrowth in the bone but stimulates the growth of breast cancer cells once they arrive in the lung.

### 3.4. Anti-IL-1 Treatment Increased the Trabecular Bone Volume of Mice

IL1β inhibition has previously been shown to inhibit bone turnover, and this is a possible mechanism by which inhibiting this pathway prevents metastatic outgrowth in the bone [2,12]. We, therefore, looked at the effects of small molecule inhibitors of IL-1 signalling on bone by assessment of trabecular bone volume using microcomputed tomography imaging as well as osteoclast and osteoblast activity by measuring TRAP5b and P1NP, respectively, in the serum. For mouse models in which E0771 cells had been disseminated into metastatic organs via intracardiac injection and received 13 days consecutive daily treatments with MLX01, VX765 or Anakinra, or 15 days consecutive daily treatments with Pacritinib increased trabecular bone volume (%BV/TV) was observed compared to mice in the control group (50 mg/kg MLX01 i.p, *p* = 0.0239; 100 mg/kg MLX01 oral gavage, *p* = 0.0197; 50 mg/kg VX765 i.p, *p* = 0.0268; 1 mg/kg Anakinra i.p, *p* = 0.0103 and 50 mg/kg Pacritinib i.p, *p* = 0.0181) (Figure 5A,B and Appendix A) (The %BV/TV of tumour or non-tumour bearing tibias is shown in Appendix A, respectively). Moreover, VX765 treatment also significantly increased trabecular bone volume (%BV/TV) of the tibia in mice orthotopically engrafted with E0771, compared with control (*p* = 0.0098) (Figure 5C) (The %BV/TV of tumour or non-tumour bearing tibias is shown in Figure 5E,F, respectively). However, significant differences were not found in osteoclast or osteoblast activity in any treatment group compared with control (Figure 5D,E). This result suggests IL-1 inhibition by MLX01, VX765, and Anakinra may inhibit tumour-cell-induced osteoclastic bone resorption as seen at earlier timepoints, as has previously been shown following administration of Anakinra [12], helping prevent the vicious cycle of bone metastasis [22], and reducing bone turnover/osteoblast genesis resulting in decreased tumour cell growth in the bone.

### 3.5. MLX01 Reduced Systemic T Cells Including CD4+ and CD8+ T Cells

Inhibition of IL1β by Canakinumab has previously been shown to adversely affect immunity in cancer patients [11]. Therefore, we used flow cytometry to analyse systemic immune cells from mouse spleens to verify whether the compounds tested affected immunity and to assess suitability for future use in clinical trials for patients with breast cancer bone metastasis (Figure 3A). Compared with control mice, MLX01 treatment decreased T cells (*p* = 0.0328), with a trend towards reduced CD4+ (*p* = 0.0823) and significant reductions in CD8+ T cells (*p* = 0.0468) (Figure 6). This result was replicated when mice received 100mg/kg MLX01 orally (*p* = 0.0051 for T cells; *p* = 0.0276 or CD4+ T cells; *p* = 0.0011 for CD8+ T cells) (Appendix A). In contrast, Anakinra did not alter CD4+ T cells but reduced CD8+ T cells (*p* = 0.0395) and monocytes (*p* = 0.0192), whereas VX765 only reduced numbers of monocytes (*p* = 0.0049) (Figure 6). Importantly our data suggest that inhibiting Caspase-1 with VX765 has the most potent anti-cancer effects, inhibiting bone metastasis whilst having the least detrimental effect on systemic immunity, suggesting that this drug may be the best option for future development.

## 4. Discussion

Our study shows that the small-molecule inhibitor that targets Caspase-1 has promising anti-tumour effects, inhibiting breast cancer bone metastasis: preventing metastatic outgrowth in the bone and reducing spontaneous metastasis to this site to the same extent or better than the IL1R antagonist, Anakinra (Figure 3E and Figure 4E), whilst having less impact on immunity (Figure 6). Moreover, although the IL1β inhibitor MLX01 only slightly reduces breast cancer metastasis to the bone, it significantly delays the onset of bone metastasis and reduces tumour growth at the primary site (Figure 3E, Figure 4E and Appendix A). In contrast, inhibiting IRAK1 by Pacritinib had small cytotoxic effects in vitro but did not reduce migration or invasion and also failed to inhibit the development of bone metastases in vivo (Figure 3L).

The expression of the IL-1-related genes, such as *Il1B, CASP1* and *IRAK1*, have been identified to be significantly up-regulated in breast cancer compared with normal mammary tissue [23,24]. Meanwhile, high expression of IL1β, IRAK1 and Caspase-1 is closely correlated with reduced overall survival and distant recurrence of breast cancer [3,23]. Especially, a significant correlation between the expression of IL1β and bone metastasis has been reported: 37% of patients with IL1β-positive primary tumours developed bone metastasis compared with 5% of patients with IL1β-negative primary tumours [3]. Therefore, targeting these molecules may provide an effective strategy for treating breast cancer bone metastasis. Our previous studies have shown that antibody-mediated IL1β inhibition or genetic ablation of IL1β in mice results in significantly reduced bone metastasis but increased growth of primary breast tumours in xenograft and allograft models, respectively [2,13]. An opposite result was demonstrated by Kiss et al., however, who showed delayed primary tumour growth in IL1β−/− (C57BL/6J background) mice after orthotopic injection with E0771 cells [25]. Similarly, we have previously found that inhibiting IL1R1 with Anakinra reduces subcutaneous tumour growth in BALB/c nude mice, but the same dose did not reduce primary tumour growth in NOD SCID mice [2,12]. Other groups have found that knocking out Nlrp3 and Casp1 in mice significantly reduces the growth of primary breast tumours and lung metastases compared with wild-type mice [26]. In addition, knockdown of IRAK1 in MDA-MB-231 tumour cells has been shown to reduce primary tumour growth in mouse mammary fat pads and subsequent lung metastasis in NOD/SCID mice [23]. These studies suggest that although IL1β seems to have contrasting effects on the primary tumour, possibly dependent on the immune status of the mouse model used and the stage at which tumour cells are in when denied exposure to IL-1, low activity of IL-1/NFKB is consistently associated with significantly reduced metastasis, especially to the bone.

There are increasing numbers of studies highlighting that small molecule inhibitors of IL-1 may be promising options for the treatment of breast cancer bone metastasis. A small molecule inhibitor of IKK (a downstream target of IL-1) reduced metastases from disseminated MDA-MB-231 cells in the bone by 58% [27]. Bishop et al. also demonstrated that an IKKe/TRAF6 (downstream target of IL-1) inhibitor could reduce osteoclastic bone destruction from 4T1 tumours growing in the bone; however, this treatment did not significantly reduce the percentage of mice that developed bone metastases [28,29]. Taking our new data together with these studies and previous studies showing near elimination of bone metastasis with Canakinumab and Anakinra, inhibiting IL1β or preventing IL1β binding to its receptor IL1R appears to almost prevent metastatic outgrowth of breast cancer in the bone [2,9,12,13]. However, inhibiting downstream targets of IL-1 (IRAK1, TRAF6, IKK) appears to be less effective, only slightly reducing the percentage of mice that develop the metastatic bone disease but reducing osteoclastic bone destruction. Therefore, we speculate that inhibiting downstream targets of IL-1 (IRAK1, TRAF6, IKK) has effects on reducing osteoclastic bone resorption, while IL1β inhibition not only reduces osteoclastic bone resorption but also blocks breast cancer metastasis to bone. In agreement with this, we have previously shown that inhibiting IL1β prevents metastatic outgrowth in the bone via several mechanisms. Firstly, inhibiting IL1β prevents wnt activation of cancer stem cells in the bone, holding tumour cells in a dormant state in this organ [9] which might explain why IL1β inhibition is more effective at inhibiting metastatic outgrowth than reducing the growth of overt metastases. Secondly, reducing IL1β in the bone inhibits expansion of the bone metastatic niche (osteoblasts, haematopoietic stem cells and endovascular cells [2]), which results in tumour cells being held in this site, preventing proliferation into overt metastases. Our μCT data from the current study show increased bone volume in mice following administration of MLX01, VX765, Anakinra and Pacritinib, indicating that these drugs all inhibit bone resorption and osteolysis (Figure 5A,B and Appendix A). In our current study, we found that 13–15 consecutive daily treatments of these drugs had no significant effects on osteoblast or osteoclast activity (Figure 5D,E). A likely reason for this is that osteoclast/osteoblast activities following intervention with drugs such as Anakinra or other factors that alter bone turnover, including ovariectomy, are commonly seen within a few hours/days, and these then normalise over time [13,30]. Therefore, 13–15 days after commencement of treatment may be too late to see the primary effects on these bone-resorbing and forming cells; however, the increased bone volume suggests that these IL-1 targeted drugs cause a net reduction in osteolysis, as previously reported for Anakinra [13].

In line with IL1β inhibition holding tumours in a dormant state, we found that inhibition of IL1 was effective in reducing bone metastasis or delaying the development of bone metastasis; however, once overt metastases had developed, IL-1 inhibitors were no longer able to reduce tumour growth (Figure 3F and Appendix A). IL1β is a pro-inflammatory cytokine, and inhibiting this molecule has been shown to prevent the infiltration of immune cells into primary tumours, facilitating their growth [13]. In our models, none of the inhibitors used increased primary tumour growth; in contrast, MLX01 reduced primary tumour growth. However, since VX765 and Anakinra both stimulated the growth of metastatic tumour cells in the lung, we hypothesise that increased tumour growth at this site may be due to a reduced immune response. Our reasoning for this is that we have previously demonstrated that IL1β has opposing effects on tumour growth in the bone compared with soft tissue [13] and that these differences in pro/anti-tumour effects are driven by differential, site-specific, immunological responses [13]. It has been postulated that IL1β promotes the recruitment of anti-tumour neutrophils to the lung metastatic niche, inhibiting metastasis to this organ. Once tumour cells are disseminated in soft tissues, inhibiting IL1β signalling with Anakinra has been shown to reduce myeloid cell accumulation in E0771 soft tissue tumours [16]. IL1β has also been shown to promote infiltration of monocytes and macrophage differentiation in soft tissue tumours [13,31]. In addition, in both melanoma and E0771 breast cancer models, IL1β has been shown to increase the anti-tumour potential of T helper (Th1) cells, and genetic knockout of IL1β in mice reduces cytotoxic CD8+ T cells in soft tissue tumours [13,18]. In contrast, IL1β appears to play a tumour-supporting function in the bone, and it has previously been demonstrated that IL1β controls myeloid cell number, reduces recruitment of anti-tumour neutrophils to bone and drives mobilisation of immune cells out of the bone marrow [13]. Whilst there is increasing evidence suggesting differential roles of IL1β in immune cell regulation of tumour growth in the bone and soft tissue microenvironments, suitable samples for further analysis of this hypothesis using the small molecule inhibitors tested in the current study were not taken.

Previous studies have demonstrated that inhibiting IL1β significantly reduced breast cancer bone metastasis, but the administration of biological agents such as Canakinumab has severe adverse effects on the immune system of cancer patients [11]. Therefore, the primary aim of our study was to identify a suitable small molecule inhibitor of the IL-1 signalling pathway that would reduce bone metastasis without seriously impacting immunity. Profiling of immune cells isolated from the spleen demonstrated that similar to previously tested inhibitors of IL-1, MLX01 reduced systemic immune cells, especially cytotoxic CD8+T cells. Whilst this may be the reason we saw a trend towards an increased percentage of mice with lung metastases, additional immune profiling of the lung tissue would be required to confirm this idea. However, despite reduced systemic immunity, MLX01 treatment delayed the growth of primary tumours; this may be because IL1β has been shown to drive both anti-tumour and pro-tumour effects via stimulating infiltration of innate anti-tumour immune cells and creating an immuno-suppressive microenvironment by increasing the infiltration of myeloid-derived suppressor cells (MDSCs), [10,32,33]. Additional experiments would be required to confirm the apparent anti-tumour effects of this drug; however, its limited anti-tumour capacity and heightened adverse effects on the immune system suggest that this may not be the optimal treatment to take forward. In accordance with the published literature, administration of Anakinra suppressed both cytotoxic T cells and monocytes, potentially explaining why when tumours did grow following treatment with this drug, they grew significantly larger than tumours treated with PBS, MLX01 or VX765 [13]. Importantly for our study, the small molecule inhibitor that was most effective at inhibiting bone metastasis, VX765, had the least effects on systemic immunity, slightly reducing monocytes, suggesting that this drug may be the best option for moving forwards into clinical trials for the prevention of future relapse in the bone.

To date, data from this current study and our previous studies [2,13] suggest that inhibiting IL-1 signalling is extremely effaceable against bone metastasis, inhibiting metastatic outgrowth and slowing the development of tumours in the bone; however, the same treatments appear to have the potential to increase metastatic outgrowth of tumour cells disseminated in lungs. It must be noted, however, that lung metastasis from mammary cancer is significantly more prevalent in syngeneic mouse models compared with human patients; therefore, increased lung metastasis observed in mice may not translate into humans [34,35,36]. Despite relatively low lung metastasis from breast cancer in humans, care must be taken not to stimulate this. In breast cancer patients, tumour cell dissemination into metastatic organs is believed to be an early event, often occurring before the diagnosis of the primary disease [37]. Therefore, whilst preventing the outgrowth of tumour cells disseminated in the bone is desirable, it is important that this is not performed at the expense of increasing metastatic outgrowth in the lung. Emerging studies suggest that this problem can be overcome by combining anti-IL-1 treatments with immune stimulatory chemotherapies and/or immune checkpoint inhibitors [13,31]. The reason for this is that IL1β is believed to exert its anti-tumour effects in the bone by inhibiting the expansion of the metastatic niche, bone resorption and the vicious cycle of bone metastasis, whereas IL1β promotes lung metastasis via reducing anti-tumour immune response [2,3,7,10,13]. Therefore, giving an anti-IL-1 treatment along with an immune stimulatory drug enables the IL-1 targeted therapy to exert anti-tumour effects in the bone without adversely affecting tumour cells disseminated at other sites. There is emerging evidence that this idea may be true; previous studies have demonstrated that combining Anakinra and the immune stimulatory chemotherapy, doxorubicin, reduces primary tumour growth and almost eliminates bone metastasis by exerting opposing effects on immune regulation [13], and a separate study demonstrated that combining Anakinra with PD-1 inhibitors resulted in significant, synergistic, anti-cancer effects in a mouse model of soft tissue tumour growth [31]. Our new data suggest that targeting IL-1 signalling upstream at Caspase-1 with VX765 is even more potent at reducing bone metastasis than Anakinra; we, therefore, suggest further experiments focused on combining VX765 with immune stimulatory therapies before considering future clinical trials.

## 5. Conclusions

In conclusion, our data show that small molecule inhibitors of IL1β and Caspase-1, MLX01 and VX765, are attractive therapeutic strategies for preventing breast cancer-induced bone disease. Previous studies have suggested that combining Anakinra with immune stimulatory chemotherapies such as doxorubicin or PD-1 inhibitors can overcome the pro-tumorigenic immune effects, significantly increasing efficacy [13,31]. Therefore, we hypothesise that combining VX765 or MLX01 with immune stimulatory drugs or targeting these small molecule inhibitors in the bone will optimise their therapeutic effectiveness.

## Figures and Tables

**Figure 1 cancers-14-04816-f001:**
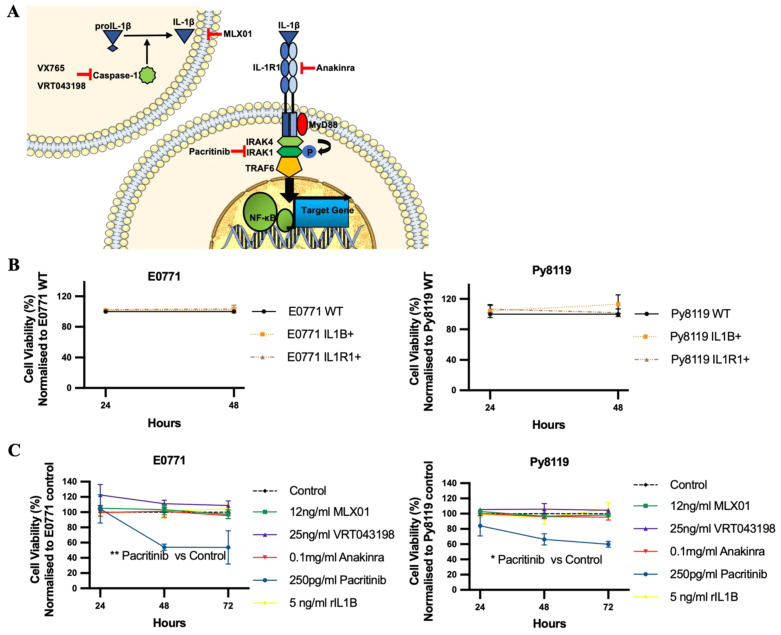
The effects of IL1β overexpression and inhibitors of the IL1β pathway on viability of E0771 and breast cancer cells in vitro. (**A**) Cartoon showing where VX765/VRT043198 (Caspase-1 inhibitors), MLX01 (IL1β secretion inhibitor), Anakinra (IL1R antagonist) and Pacritinib (IRAK1 inhibitor) inhibit the IL1β signalling pathway. (**B**,**C**) Effects of overexpression of IL1β (IL1B+), overexpression of IL1R1 (IL1R1+), inhibition of IL1β (12 ng/mL MLX01), inhibition of Caspase-1 (25 ng/mL VRT043198), inhibition of IL1R (0.1mg/kg Anakinra) and inhibition of IRAK1 (250 pg/mL Pacritinib) and recombinant IL1β (5 ng/mL) on tumour cell viability, normalised to control. All experiments are from three biological experiments and their graphs represent mean + SD, * *p* < 0.05, ** *p* < 0.01.

**Figure 2 cancers-14-04816-f002:**
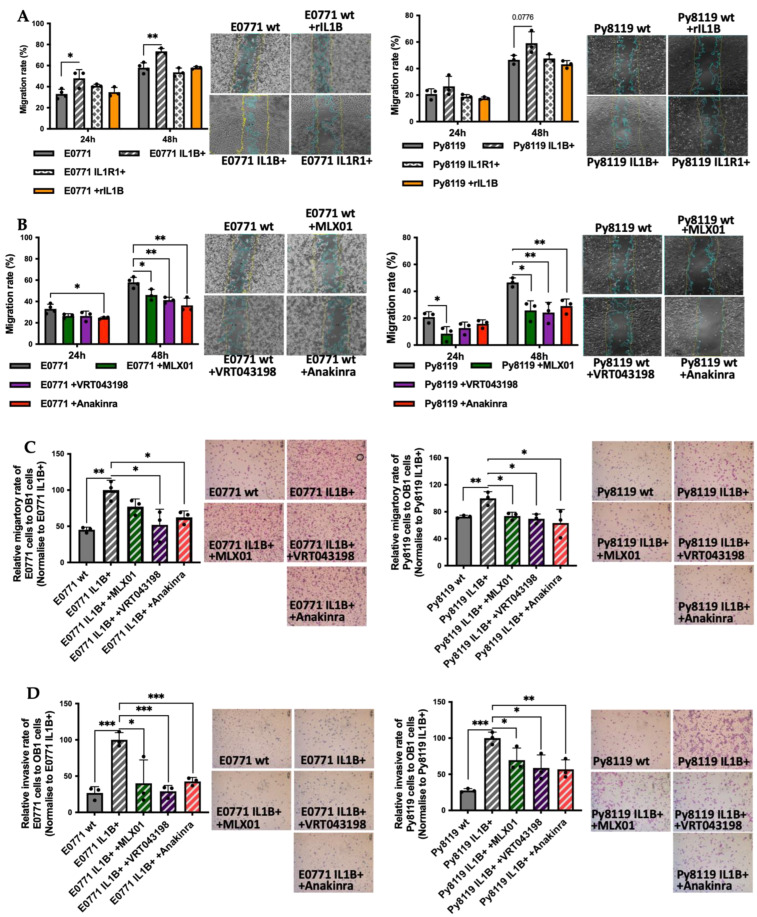
The effects of IL1β overexpression and inhibitors of the IL1β pathway on migration and invasion of E0771 and Py8119 breast cancer cells in vitro. (**A**,**B**) Effects of overexpression of IL1β (IL1B+), overexpression of IL1R1 (IL1R1+), inhibition of IL1β (12ng/mL MLX01), inhibition of Caspase-1 (25 ng/mL VRT043198), inhibition of IL1R (0.1 mg/kg Anakinra) on cell migration of wide type E0771 and Py8119 cells. Data are presented as the percent of scratch closure following 24 h and 48 h in culture, normalised for the beginning of scratch. Representative images of cells at 48 h are shown where the yellow and blue lines are scratch edges at 0 h and 48 h, respectively. (**C**,**D**) The effects of alteration of IL1β signalling on migration and invasion towards pre-osteoblast for 24 h (OB1 cells). All experiments are from three biological experiments and their graphs represent mean + SD, * *p* < 0.05, ** *p* < 0.01, *** *p* < 0.001.

**Figure 3 cancers-14-04816-f003:**
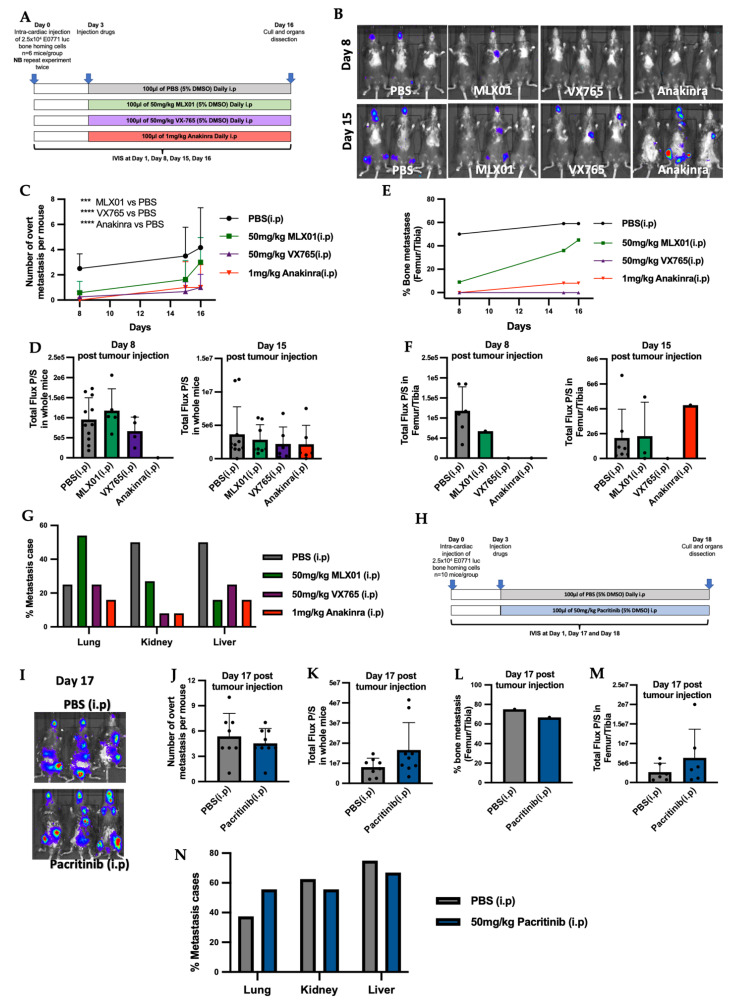
Inhibiting IL-1 signalling with IL1β inhibitor MLX01, Caspase-1 inhibitor VX765 and IL1R antagonist Anakinra reduced metastatic outgrowth from E0771 cells disseminated in the bone. (**A**) Outline of experimental protocol for administration of PBS (control), MLX01, VX765 and Anakinra in mice following intra-cardiac injection of E0771. (**B**) Representative whole body IVIS images of mice at Day 8 and Day 15 post injection of E0771 cells. (**C**) Numbers of overt metastases per mouse 8-, 15- and 16-days post injection of E0771 cells following daily administration of PBS (control), MLX01, VX765 or Anakinra. (**D**) Total luciferase expression (photons per second (P/S)) of whole mice 8 and 15 days after E0771 cell injection. (**E**) Numbers of mice that developed bone metastases 8, 15 and 16 days after E0771 cells had been disseminated in the bone following daily administration of PBS (control) MLX01, VX765 or Anakinra. (**F**) Size of tumours that grew in the bone as assessed by photons per second (P/S) in luciferase expressing E0771 tumour cells. (**G**) Shows effects of PBS (control), MLX01, VX765 or Anakinra on metastatic outgrowth of tumours following dissemination into the lungs, kidneys and livers. (**H**) Protocol outline for administration of Pacritinib in mice after intra-cardiac injection of E0771. (**I**) Representative images of whole body IVIS imaging 17 days post E0771 injection. (**J**) Numbers of overt metastases per mouse 17 days post injection of E0771 cells and following daily administration of PBS (control), or Pacritinib. (**K**) Total luciferase signal from E0771 tumour cells (photons per second (P/S)) in whole mice 17 days following injection of E0771 cells and administration of PBS (control) or Pacritinib. (**L**) Numbers of mice that developed bone metastases 17 days after E0771 cells had been disseminated in the bone following daily administration of PBS (control) or Pacritinib and size of tumours that grew in the bone as assessed by IVIS imaging in (**M**). (**N**) Shows effects of PBS (control), or Pacritinib on metastatic outgrowth of tumours following dissemination into the lungs, kidneys and livers. *** *p* < 0.001, **** *p* < 0.0001.

**Figure 4 cancers-14-04816-f004:**
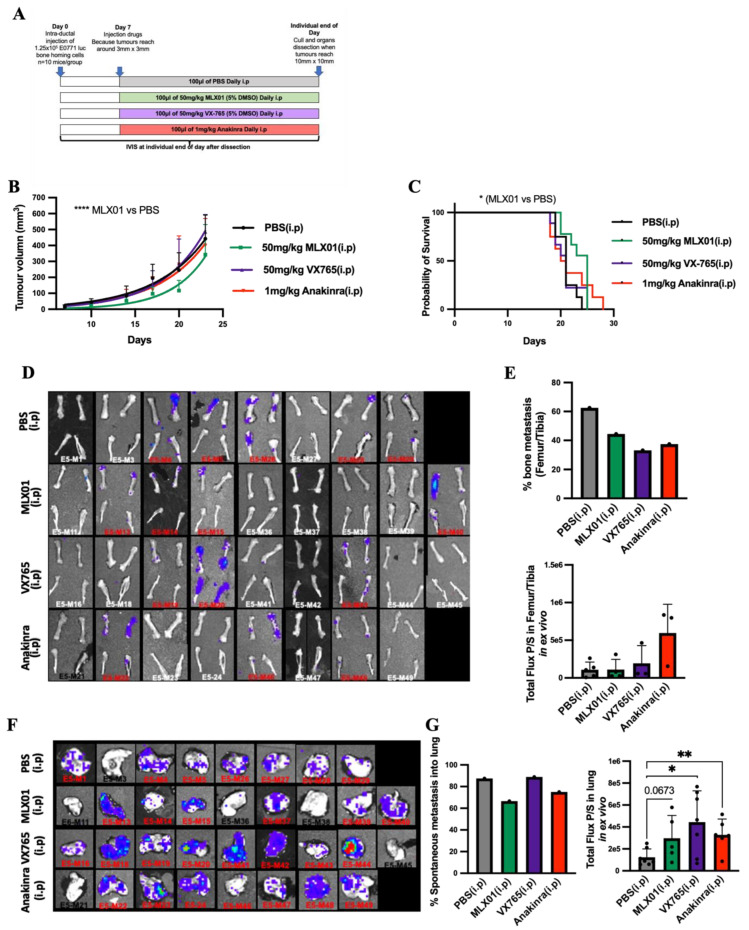
The effect of inhibiting signalling of the IL-1 pathway on growth of E0771 cells in mammary glands and spontaneous metastasis to lung and bone (**A**) Protocol outline for orthotopic injection of E0771 cells into mammary glands and administration of PBS (control), MLX01, VX765 or Anakinra in mice. (**B**) Primary tumour volume following daily administration of PBS (control), MLX01, VX765 or Anakinra. (**C**) Survival curve (**D**) IVIS images of dissected femur/tibia ex vivo following exposure to D-luciferin for 5 min. (**E**) Percentage of mice that developed spontaneous bone metastasis and size of tumours in femurs/tibiae as assessed ex vivo by IVIS. (**F**) IVIS images of dissected lungs, ex vivo (**G**) Percentage of mice that developed spontaneous lung metastasis and size of tumours in lungs as assessed by IVIS, ex vivo. The graphs represent mean ± SD, * *p* < 0.05, ** *p* < 0.01, **** *p* < 0.0001.

**Figure 5 cancers-14-04816-f005:**
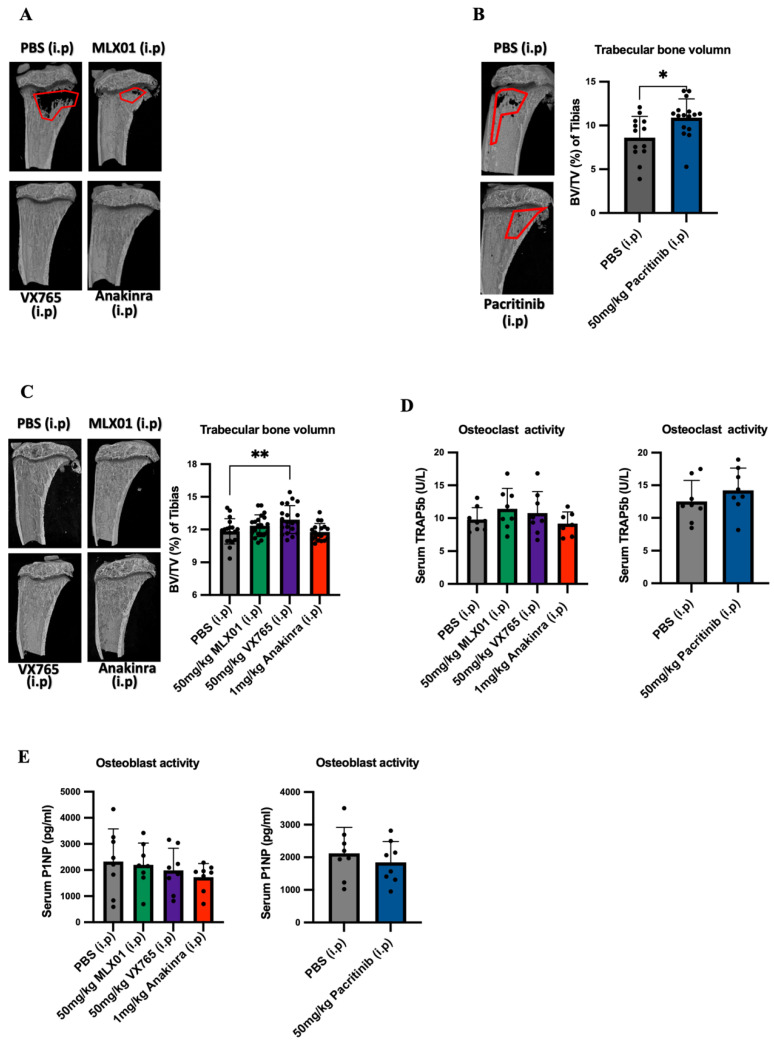
Effects of inhibiting IL1β, IL1R1, Caspase-1 and IRAK1 on bone turnover and volume (**A**) Bone volume/tissue volume (BV/TV) % of trabecular bone in the mouse tibia following dissemination of E0771 cells in this site and 13 consecutive daily treatments with PBS (control), MLX01, VX765 or Anakinra (Study Figure 3A) or (**B**) 15 days 15 consecutive daily treatments with Pacritinib (Study Figure 3H). (**C**) Bone volume/tissue volume (BV/TV) % of trabecular bone in mouse tibia following orthotopic injection of E0771 and consecutive daily treatments with PBS (control), MLX01, VX765 or Anakinra until primary tumour reached 1000 mm^3^. (**D**,**E**) ELISA analysis of TRAP5b (**D**) or P1NP (**E**) in mouse serum following dissemination of E0771 cells and 13 consecutive daily treatments with MLX01, VX765 or Anakinra (Study Figure 3A) or Pacritinib (Study Figure 3H). Graphs represent mean + SD, * *p* < 0.05, ** *p* < 0.01.

**Figure 6 cancers-14-04816-f006:**
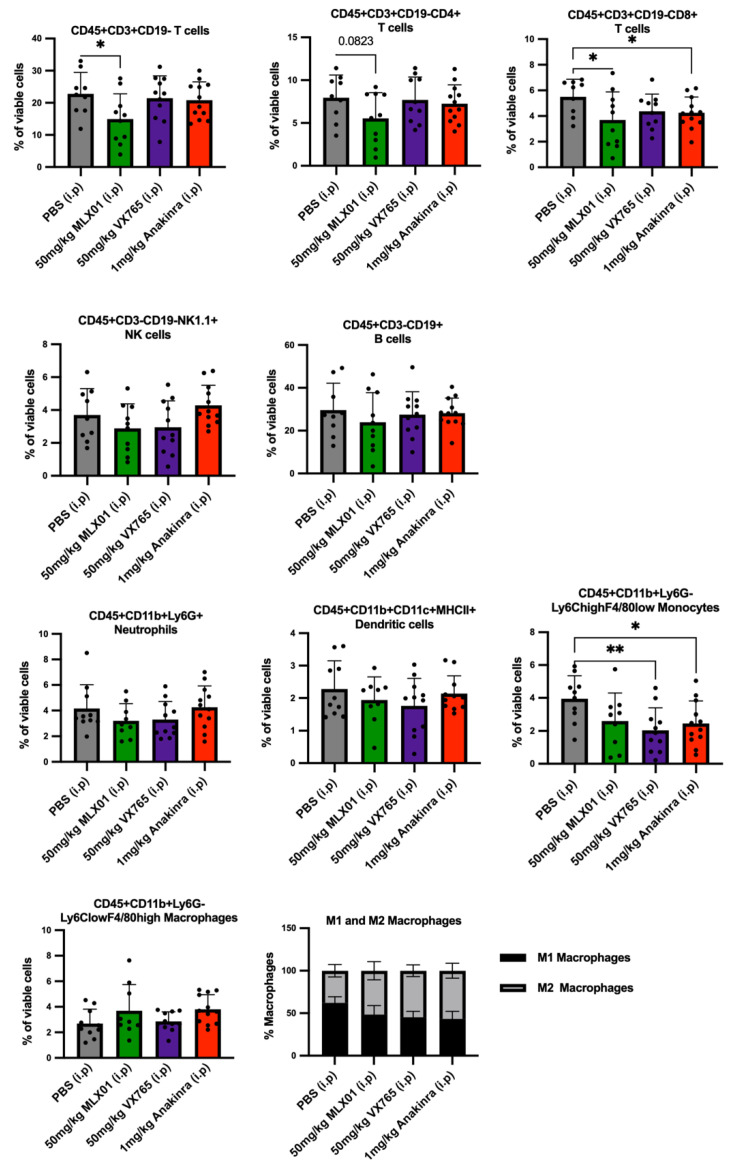
Effects of MLX01, VX765 and Anakinra on systemic immune cell populations. Immune cell populations: T cells (CD45+CD3+CD19−); CD4 T cells (CD45+CD3+CD19−CD4+CD8-); CD8 T cells (CD45+CD3+CD19-CD4-CD8+); B cells (CD45+CD3−CD19+); Natural killer cells (CD45+CD19-NK1.1+); Neutrophils (CD45+CD11b+Ly6G+); Dendritic cells (CD45+CD11b+CD11c+MHCII+); Monocytes (CD45+CD11b+Ly6G-Ly6ChighF4/80low) and Macrophage (CD45+CD11b+Ly6G-Ly6ClowF4/80high); M1 Macrophage (CD45+CD11b+Ly6G-Ly6ClowF4/80highMRC1−); M2 Macrophage (CD45+CD11b+Ly6G-Ly6ClowF4/80highMRC1+) were analysed from dissociated spleen of C57BL/6 mice bearing E0771 tumours following 13 consecutive daily treatments with PBS (control), MLX01, VX765 or Anakinra (Study Figure 3A). Cell populations were quantified by flow cytometry and gating strategy shown in Appendix A. Data represent mean + SD, * *p* < 0.05, ** *p* < 0.01.

## Data Availability

The Data can be shared up on request.

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
