# Peer review of "Novel Methods of Targeting IL-1 Signalling for the Treatment of Breast Cancer Bone Metastasis"

_cancers, 2022, doi:10.3390/cancers14194816_

Round 1
Reviewer 1 Report (Previous Reviewer 1)
The authors have adequately addressed all critiques and provided rationale for experimental design choices in the manuscript. The manuscript has been sufficiently improved for acceptance.
Author Response
We would like to thank the reviewer for the time they have put into looking at our work and their positive interpretation of our manuscript.
Reviewer 2 Report (Previous Reviewer 3)
I have considered attached authors' reply to my concerns. I can see that authors added some significant data to address the lack of any information on IL-1b inhibition by MLX01.
However, they did not address this fully, because the original manuscript describes multiple experiments in which MLX01 was delivered orally while in their response they have evaluated in vivo on-target effect only upon i.p. delivery (apparently, since the dose 50mg/kg corresponds to this route in the manuscript). Without addressing bioavailability of MLX01, interpretation of the data obtained upon oral delivery in the original manuscript is not possible.
Additionally, in accordance to their new data, MLX01 is not a direct inhibitor of IL-1b, but instead it is an inhibitor of its secretion/production. Therefore it should be presented and discussed as such. Additionally, authors should ensure that they have included a proper control demonstrating MLX01 selectivity towards IL-1b.
Author Response
We would like to thank the reviewers for their helpful comments we have addressed these concerns in the following way.
- We show data from two models in which mice have been treated with MLX01. In the first model 50mg/kg MLX01 is administered via i.p injection, in the second model 100ug/kg is administered by oral gavage. We have provided additional data to show that these doses and rates of administration result in the same (50%) reduction in IL-1B section (supplementary figures 2B and 2C).
- We are in agreement with the reviewers that MLX01 does not alter transcription of IL-1B but, instead alters secretion. We have shown additional data in supplementary figure 2C to confirm this and adjusted the text accordingly.
Reviewer 3 Report (Previous Reviewer 4)
The authors answered all my concerns. But I noticed that I missed one point about MLX01 when I read other reviewers' comments.
Although authors provided some evidence showing MLX01 inhibited the IL1b production, it is not clear how it works or its direct target. Is it directly suppresses the transcription of IL1b or indirectly inhibits via targeting other targets? Therefore, more evidence is required for its inhibiting mechanism. Or some testing results from the origin of this inhibitor may help.
Author Response
We have included additional data to show that MLX01 does not alter transcription of IL-IB (supplementary figure 2D). Furthermore we have shown that that intra-peritoneal injection of 50mg/kg MLX01 and oral gavage of 100mg/kg MLX01 both inhibit secretion of IL-1B by 50%. Additional data relating to the mechanisms of action of this drug are currently being patented.
Round 2
Reviewer 2 Report (Previous Reviewer 3)
In the revised manuscript, the authors provided a sufficient body of data proving specific targeting of IL-1beta secretion by MLX01 both in vitro and in vivo and upon delivery by the routes used within the study. Taken together with the extensive revision and response to the points raised by other reviewers, the manuscript is surely significantly improved and can be recommended for acceptance after addressing the following minor issues:
- in the abstract, it should be clearly stated MLX01 (IL1β secretion inhibitor) to avoid confustion
- in the Figure 1A MLX01 is still shown as a direct inhibitor of IL1beta, it should be placed correctly in accordance with the conclusions from the data
- in the Figure S2B, the symbol for statistical significance is missing
- in the FIgure S2 legend the lab slang "tumour IVISed size" should be replaced with the suitable scientific terminology of authors' choice
Author Response
We would like to thank the reviewer for their helpful comments. We have addressed these as follows:
1. We have clearly stated in the abstract that MLX01 is an inhibitor of IL-1B secretion (line 28).
2. We have updated Figure 1A MLX01 to indicate that MLX01 inhibits secretion of IL-1B and edited the figure legend accordingly.
3. We have included the symbol for statistical significance in Figure S2B.
4. In the Figure S2 legend the lab slang "tumour IVISed size" has been replaced with "normalised to total flux photons/second"
Reviewer 3 Report (Previous Reviewer 4)
Authors answered all my concerns.
Author Response
We would like to thank the reviewers for their appreciation of our manuscript and taking the time to look at this.
This manuscript is a resubmission of an earlier submission. The following is a list of the peer review reports and author responses from that submission.
Round 1
Reviewer 1 Report
The manuscript entitled “Novel methods of targeting IL-1 signaling for the treatment of breast cancer bone metastases” investigates the role of IL-1b, IL1R1, and downstream mediator IRAK1 using available pharmaceutical agents to determine optimal pathway inhibition for metastatic control. The authors provide a nice overview of the current concerns of IL-1 therapeutic intervention (such as the contradictory inhibition seen in bone and soft tissues) and provide a rationale for inhibition of the signaling cascade at various locations of the pathway. The authors provide in vitro and in vivo analysis of these compounds. In addition, systemic immune profiling is performed to determine alterations due to treatment with the compounds of interest. Overall, the manuscript is well written with sufficient description of the methods employed and results obtained.
Major Concerns:
1. The authors investigate transwell migration and invasion in the presence of OB1 cells to suggest the bone specific nature of the interactions observed. Did the authors investigate other bone cells (eg osteoclasts- differentiated RAW cells) or other tissue types (lung, liver, kidney) since they were studied in the paper? Differences in migration and invasion towards different attractant targets would be important indicators of the suggested specificity associated with OB1. These additional experiments would be strongly supportive of the bone specific nature of the authors other observations.
Minor concerns:
1. Throughout the manuscript, significant differences are not clearly denoted.
a. For example, in figure 1, differences are noted in the manuscript text (lines 218-219) but no apparent statistical difference notation is present in the figure.
b. In figure 2, panels A and B have asterisks or p-values, but not panels C/D.
c. Figure 3 contains a legend for significant differences, but there are no indicators on any of the panels.
d. The above is not an exhaustive list. Therefore, the authors should clearly, and consistently, update statistical difference notation throughout the paper.
2. It does not appear that supplemental figure 6 is referenced in the main manuscript.
3. In section 3.1.5 the authors suggest that the systemic immunologic alterations they observed from cryopreserved spleen “suggests that increased lung metastasis may be associated with reduced immunity.” This statement appears to be conjectural since the authors are looking at systemic immunity rather than the immune environment present in tumor bearing lung. In addition, the systemic immune changes observed would need to be equally applied to bone in this context. While systemic immunity is a fine measure for these experiments, the authors must also consider the importance of the tumor/metastatic microenvironment in a tissue specific context as it is likely having a more direct effect on localized tumor growth.
a. This sentiment is echoed in the discussion (lines 565-570) with the authors speculating that VX765 having a benefit simply based on systemic immunity.
b. A brief discussion pursuant to the above would be sufficient.
4. While the differences the authors observed in soft tissue vs bone metastases are interesting, the authors should also describe the context of suggested treatment. Many patients present with sub-clinical and micro-metastatic disease. Since this disease may not be clinically evaluated, the treatments proffered within may exacerbate growth of these soft-tissue metastases at the expense of bone metastases. It is plausible that there is a patient stratification context where this would be feasible, but the authors do not discuss these complications.
a. The authors are encouraged to discuss the potential effects of these treatments on patients with sub-clinical non-bone metastases in the context of their observations.
5. In the conclusion (section 5) the authors make a speculative conclusion concerning the combination of VX765 or MLX01 with PD-L1 or similar immunomodulators. This statement should be moved, or at least initiated in the discussion, to provide context for this conclusion.
a. In addition, the authors should provide rationale for the addition of immune modulators after attempting to inhibition one of the most pro-inflammatory cytokines.
6. In the supplementary figure 4 legend, there are references to figures 2A and 2H. Should these be figure 3?
Reviewer 2 Report
Here the authors test a hypothesis that targeting of pro-inflammatory cytokine IL-1beta signaling could be an effective strategy to reduce breast cancer bone metastasis without adversely affecting primary tumor growth as well as immune cell signaling and functions. Authors utilize syngeneic mouse mammary tumor model using E0771 and Py8119 breast cancer cells in conjunction with various inhibitors of IL-1 signaling. Specifically, authors utilize IL1beta inhibitor MLX01, Caspase-1 inhibitor (VRT043198; VX765), IL-1R antagonist Anakinra, and IRAK1 inhibitor Pacritinib. In vitro studies indicated that none of the inhibitors caused inhibition of cell viability/growth, but inhibited migration and invasion aspects of the murine breast cancer cells. In the vivo studies Luc2-GFP-expressing E0771 murine breast cancer cells were grown as orthotopic tumors either by intra-cardiac injections for disseminated metastases or by intra-ductal injections for spontaneous tumor growth, and the animals were treated with respective agent through intra-peritoneal injections or by oral gavage. The outcomes consisted of monitoring tumor metastases in bone (Tibia), in lungs, and changes in trabecular bone volume as well as immune cell populations in spleens. The studies are generally well performed with appropriate statistical analyses where applicable. The rationale for targeting IL-1 signaling is reasonable as high expression of IL-1beta, IRAK1, and Caspase-1 correlate with reduced overall survival and distant breast cancer metastases. Moreover, anti-IL1beta antibody (Canakinumab) or the IL1R antagonist (Anakinra) inhibit breast cancer bone metastases but promote growth of primary breast tumors in preclinical models. Although these inhibitors are approved for clinical use by USFDA/NICE, they have poor oral bioavailability and are often administered sub-cutaneously. A large body of figures and supplementary data generally are supportive of the conclusions. However, the studies involving IL1beta inhibition by MLX01, IL1R antagonist Anakinra, or IRAK1 inhibitor Pacritinib are in large part confirmatory of prior studies. Nevertheless, the potential of caspase-1 inhibitor VX765 to prevent metastatic outgrowth of breast cancer to bone is interesting. Although caspase-1 targeting affects spontaneous bone metastasis by breast cancer cells that is similar to IL-1R antagonism, the caspase-1 inhibition as well as targeting of IL-1beta/IL1R caused increased lung metastasis and metastatic outgrowth of breast cancer cells to lungs. Moreover, although targeting of IL-1beta signaling does prevent initial metastatic outgrowth of breast cancer cells to bone, it does not inhibit growth of established bone tumors. The studies showing inhibition of bone cancer cell induced osteoclastic bone resorption following IL-1 inhibition are essentially confirmatory of prior studies. Importantly, IL-1beta inhibition (MLX01) or IL1R inhibition (Anakinra) caused significant reduction in CD8+ T cells but Caspase-1 inhibitor (VX765) only reduced monocytes suggesting that reduced immune responses could permit breast cancer metastases to lungs. Here the conclusion in section 3.1.5 that caspase-1 inhibition causes potent anti-cancer effects, inhibits bone metastasis with least detrimental effects on systemic immunity appears to be an over-interpretation as VX765 does promote breast cancer metastases to lungs. Overall, the conclusions are largely confirmatory of prior findings, and although Caspase-1 inhibition may be potentially effective in preventing breast cancer outgrowth to bone, further studies will be necessary to robustly establish potential of caspase-1 targeting as an effective anti-cancer strategy particularly in combination with current immune-stimulatory modalities.
Specific Comments:
11. Please include VX765 in Abstract line 28.
22. Line 98, cultured in DMEM medium.
33. Please explain why cells were pretreated with Mitomycin C (Methods line 111).
44. Please correct cell number in line 120 (30,000 cells).
55. Please explain why drug administration was carried out by IP injections and not IV injections.
66. Line 166 ethanol for analysis….
77. Please carefully check whether results indicate correct figures (line 222, Figure S2C; line 311, Figure 3E; Line 315, Figure 3E).
88. Is panels C and D in Figure 2 are for 24h and 48h. If so, please indicate that in legend.
99. Line 300, outgrowth was compared….
110. Line 386, inhibitors tested had reduced number….
111. Line 399, pathway on growth of….Line 423, delete of tibia.
112. Line 486, IL-1 related genes. Line 550, pro-inflammatory. Line 593, Charity for providing the salary.
113. In figure 1a, and 1b, please also show representative western blots for levels of respective proteins.
114. Figure 3, please indicate p-values in the graphs too.
115. Figure 1C and 1D, correct Normalized to.
Reviewer 3 Report
The manuscript "Novel methods of targeting IL-1 signalling for the treatment of breast cancer bone metastasis" reports assessment of several new approaches to target breast cancer bone metastasis via IL-1beta signaing in an attempt to battle negative effects on primary tumor growth and immune-related adverse reactions. The authors address alternative ways of targeting the pathway via different approach to direct IL-1beta inhibition, or targeting caspase-1 and IRAK. While the study overall addresses an interesting and novel point in an overall comprehensive way, it contains a critical issue which must be addressed prior to any detailed review.
Specifically, the authors use in the study what they call IL-1b inhibitor MLX01. However, unlike all of the other compounds they have used, this one is unpublished and it is extremely surprising that authors use it throughout the entire work as a matter-of-fact IL-1b inhibitor, while providing absolutely no data on its actual capacity to do so. This peculiar situation obviously does not allow a proper review of the manuscript since the results obtained with MLX01 are so deeply integrated into it. I see only two possible solutions to this critical issue:
1) Authors should provide in a manuscript a comprehensive set of data showing that MLX01 is, as they claim, a direct small-molecule inhibitor of IL-1b capable of acting at the concentrations and doses they describe. This should demonstrate physical binding or other proof of a direct and unassisted interaction with IL-1b, dose-response inhibition in two relevant orthogonal IL-1b signaling readouts in vitro, and, since authors use oral delivery of the compound, the data demonstrating its bioavailability and on-target effect in vivo. Given the topic of the manuscript, which is based on different mechanisms of agents, nothing short of this will be sufficient.
2) Authors can completely remove the data on MLX01 from this study or resubmit it when these data will be published.
Reviewer 4 Report
In the manuscript “Novel methods of targeting IL-1 signalling for the treatment of breast cancer bone metastasis”, Zhou et al. demonstrate that pharmacological inhibition of Caspase-1, IL1β and IL1R in the IL1 signaling suppressed breast cancer bone metastasis. It has been previously reported that IL1b antibody and IL1R1 antagonist inhibit bone metastasis. And their mechanism has been previously discovered. Thus the findings in this manuscript are not very new, except having different targets in IL1 signaling. However, this manuscript is still a good supplement for studying the potential therapy by inhibiting IL1 signaling during treating breast cancer bone metastasis. This study is well designed, performed and analyzed.
I only have minor concerns.
1. The ex vivo bone metastasis analyses by intra-cardiac injection of breast cancer cells are well performed. Live imaging of the whole mouse and dissected tissues are clearly illustrated. However, some pictures of live imaging in Fig 3B show that several mice had luminescence signals only on their heart all the time. This outcome is very likely due to the failed tumor cell injection, which means that during the intra-cardiac injection, the tumor cells were injected into the heart wall, not the ventricle, causing trapped tumor cells in heart. Therefore, these mice should not be considered as having tumor cells in their circulation nor considered as developing none bone metastasis. The data of bone metastasis should be re-analyzed.
2 Another related question to Fig 3B - were these images taken and analyzed under the same range of luminescence intensities? If so, the range scale should be illustrated in the figure.
3. Fig S4B, S4D, S5I show some tumor signals on dissected lung. Normally researchers study ex vivo lung metastasis by tail vein injection. Lung metastasis is not common if the intra-cardiac injection is performed properly into the left ventricle, because tumor cells have to go through the blood vessels within the whole body, return to the right ventricle and then arrive at lung. Authors should explain these lung metastases.
4. It should be noted that lung metastasis from breast cancer is not as frequent as bone metastasis in human, which is opposite to mouse models. This should be mentioned in the main text where it describes increased spontaneous lung metastasis by inhibitors.
5. On Line 426-429, authors wrote “This result suggests IL-1 inhibition by MLX01, VX765 and Anakinra may inhibit tumour cell induced osteoclastic bone resorption as seen at earlier timepoints, as has previously been shown following administration of Anakinra [12], helping prevent the vicious cycle of bone metastasis [22].” Authors should mention another possibility here that inhibitors suppress osteoclasts at first, resulting decreased tumor cell growth in bone, although this point has been mentioned in the discussion.
6. Authors should include the full names of abbreviations when they appear for the first time in the article, for example, IRAK1.
7. In Fig 1A, caspase 1 should be an intracellular protein.
8. The time frame of drug treatment in Fig 3A should start from Day 3 according to the text. There are similar errors in other figures.
9. The figure citations should be double checked. For example, Fig S2, 4 should be cited in the main text. Fig S4A, B should be related to Fig 3A, not 2A.
